

# Development of a power station unit in a distributed hybrid acquisition system of seismic and electrical methods based on NB-IoT

Feng Guo[1], Qisheng Zhang[1], Shenghui Liu[1]

[1] School of Geophysics and Information Technology, China University of Geosciences Beijing, Beijing, 100083, China

*Correspondence to*: Qisheng Zhang (zqs@cugb.edu.cn)

**Abstract.** In this paper, we propose a new type of power station unit with wireless data transmission capability. This work breaks the limitation that conventional equipment is unable to upload data directly to central unit. Based on that, a novel distributed geophysical data acquisition architecture is also proposed. Designs that realize functions including isolated high-

power output, power management and monitoring, high-speed data transmission as well as high-precision clock synchronization etc. are introduced in this article. The prototype was packaged then, and a series of evaluation experiments were implemented to verify the key parameters of the instrument. Experiment results proved that the overall design of the instrument is feasible, and the key parameters outperforms the industry leading instrument LAUL-428. Due to the wireless networking strategy, the proposed instrument further realizes remote control and real-time data playback through the host

computer software, making it suitable for joint geophysical exploration as well as microseismic monitoring. In a sense, a new birth was also given to the conventional seismic and electrical acquisition stations.

## 1 Introduction

The ambiguity problem of a single geophysical exploration method has always been a tricky problem in geophysical prospecting (Guo et al., 2020). Multi-method joint exploration has then become popular since it remains the advantages of

different geophysical methods while reducing ambiguity problems. Among them, the joint exploration of seismic and electrical methods has become the research direction of many scholars (Garofalo et al., 2015; Wagner et al., 2019). However, the development of corresponding instrument is still at the very beginning.

Taking seismic exploration instruments as an example, the current seismometers can be divided into wired telemetry and wireless telemetry seismometers according to the communication method (Zhang et al., 2014). In the 1970s, SERCEL had

already launched the SN338 digital seismograph and corresponding single-channel acquisition station (Qiao et al., 2019), then they launched SN388, 408UL, 428XL, 508XT and other products successively. At present, the most advanced seismic acquisition system in the world is 508XT system, which has a real-time acquisition capability. The 428XL seismic acquisition system, however, is the most widely used system in the entire geophysical prospecting industry at present, who has the most representative system arrangement structure. The system shown in Figure 1(a) is mainly composed of a central

control unit (e-428), a line control interface (LCI-428), a transverse station (LAUX-428), a line power (LAUL-428) and a



collection chain (Link). The collection chain consists of several acquisition stations (FDU-428) or digital sensors (DSU-428) that connected by cables (Sercel, 2006). LCI is the interface between the arrangement and e-428 client, which plays the role of high-speed Ethernet routing. The main function of LAUX is to collect and package data from the arrangement and send them to another LAUX unit or LCI. LAUL is connected directly with Link through the cable, then power is supplied to the

acquisition nodes on the Link and the data are relayed as well (Sercel, 2007). The 508 system upgrade this structure further by combing LAUL and LCI, and replace them with CX-508 (Sercel, 2023). This, however, didn't combine wireless communication function (Duan, 2019). Besides, the Inova G3i NXT (Inova, 2022), I/O Scorpion (Gan, 2013), and Seismic Instruments (2023) shared the same system structure with 428XL, and a transverse station or line control interface is necessary. Other geophysical exploration system, such as that for Multi-channel Transient Electromagnetic Method

(MTEM), share the similar structure in cabled manner with above mentioned system, and wireless quality check function added (Dong, 2015).

In despite of the burst of nodal seismograph e.g., Hawk, SmartSolo, WTU-508 (Lv et al., 2022) and others developed by Tian (2020) etc., the wired seismograph system is suitable for certain exploration situations (Ellis, 2014). As is found by Lansley (2012), the best way of exploration system structure in the near future should be the combination of wired and

wireless system, which could also be proved by the upgrade from 428XL system to the 508XT system developed by industry leading company. In fact, as analyzed by Dean et al.(2013), for dense exploration with receiver interval less than 40m, the cabled system weights less than nodal system, and specifically, when receiver interval is 10 m, the cabled system weights only 24% of the nodal system.

Combining the distributed seismic and electrical acquisition stations developed by our team, a new type of power station unit

(PSU) is proposed, thus a geophysical instrument networking method is also proposed in this article. The point is to retain the original functions of the LAUL and integrate the upward communication and human-computer interaction capabilities of the LCI by wireless communication. Therefore a 120-channel acquisition arrangement is developed based on a single PSU. This arrangement can be further expanded through wireless networking, which could be very convenient for field geophysical exploration on complex terrain (R. G. Heath, 2008). Compared with traditional wired systems, the proposed

structure can reduce a considerable number of equipments to form an efficient work and data transmission flow.

## 2 Overall design

The proposed PSU based system is shown in Figure 1(b). A single PSU could connect and supply power to a series of acquisition stations (ACs) via the cable by two connectors, one on each side. The PSU serves as the main control unit by sending the control commands to the ACs in a relay manner. After signal acquisition, the data are relayed on the line to

upload from the AC at the end of the arrangement. The intermediate ACs add their own data to the data frame and then continue to upload forward until the PSU. Meanwhile, the PSU also performs as the human-computer interaction interface. Unlike the 428XL system, no cables or additional LCI are required between PSU and the central control unit. Instead, Wi-Fi


and Narrow Band Internet of Things (NB-IoT) are implemented as remote data transmission techniques, thus expanding the system scale to a larger extent through wireless networking. This layout is flexible and suitable for complex terrains in the

wild or urban underground space detection.

## 2.1 Design targets

The design target is to integrate the functions that belonged to LCI and LAUX originally, into the PSU. Therefore, the aims are:

1) Solving the problem that output voltage level of external battery decrease when discharging to provide stable and high-
quality power to the acquisition station arrangement. And monitoring the power status at the same time.
2) Receiving control commands from the host computer and configuring the acquisition station arrangement accordingly, while uploading received data to the host computer.
3) Buffering and preprocessing the uninterrupted data stream uploaded by the acquisition stations.
4) Providing a high-precision clock source for connected ACs and add absolute time information to the data.

## 3 Design of hardware circuits

The PSU adopts a modular hardware design, consisting of interface board, boost board (2 pieces), power board, ARM® main control board (ARM® board), FPGA (Field Programmable Gate Array) board, clock board, and built-in back-up lithium battery. The hardware structure is shown in Figure 2.

All circuit boards are fixed in the enclosure shown in Figure 3. The interface board is installed on the connector side, and the
other circuit boards are fixed together by metal screws on the bottom of the case. Between circuit boards, board-to-board connectors or flat cables are used. In this way, the spacing and quantity of the boards are adjustable, therefore the circuit boards could be replaced easily for fixing and upgrading.

### 3.1 Design of interface board

The interface board mainly realizes the transfer from each interface on the instrument enclosure to the internal circuits,
whose shape and position therefore correspond to each interface on the enclosure.

Another main function of the interface board is to select a reliable power supply from external battery or the built-in battery. In this case, the power path controller LTC4416 is introduced to control 2 sets of external P-channel Metal Oxide Semiconductor Field Effect Transistors (MOSFET) to realize a nearly ideal diode function for the power switching circuit. It also expands the function of power sequencing while realizing power supply combination and reverse connection protection.

When valid external power supply is connected, the power source would be selected according to the preset voltage threshold, thereby saving the back-up battery. If voltage level of external source drops lower than that threshold, back-up

battery would be switched to automatically to avoid the power interruption due to external power failure during field exploration.

## 3.2 Design of boost board

The function of the boost board is to output and monitor 48 V power for acquisition stations which is boosted from the unstable 12 V input from external battery. To load more stations and provide flexibility while set-up, we designed a dual output link strategy, which means two boost boards are in need for each power link.

### 3.2.1 Boost circuit

The voltage boost is realized by VI-204-CX type DC-DC (direct current-direct current) module, whose input voltage range is
10-20 V, and the output voltage is 48 V. In this case, our theoretical maximum output power is 75 W for single boosted output source, and the conversion efficiency can reach up to 90%. The power output of the PSU is controllable; therefore, the boost output can be turned off to reduce power consumption during parameter configuration when power supply to the acquisition stations is not in need.

### 3.2.2 Output power monitoring circuit

In the PSU, the output power source is monitored by measuring current and voltage, and simultaneously calculate the power value relying on LTC2945 utilizing its integrated multiplier who performs digital multiplication directly on the measured current and input voltage data to generate a 24-bit power value. Those acquired data are continuously sampled and stored in the internal register of LTC2945 and meanwhile the maximum and minimum values are updated (Linear Technology, 2012). Both maximum and minimum thresholds for voltage, current and power can be configured. When physical value beyond this
threshold range is detected, a fault signal will be output through the ALERT pin. Microcontroller Unit (MCU on power board) configures this signal as an external interrupt source, and will give rapid response to this interrupt, which is shutting down the 48V output, in case abnormal situations happen.

## 3.3 Design of power board

The main function of the power board is to monitor the input power and convert it to 5 V, 3.3 V, 1.1 V power for other
internal circuit boards. In addition, the control of the boost board, switches and LED indicators on the instrument panel are also realized by the power board.

### 3.3.1 Power board control circuit

The main task of MCU on power board mainly include as below:

**Table 1.** Power board control MCU task and hardware resources in need



| MCU hardware resources | Task |
|---|---|
| General Purpose Input/Output | Signal control |
| External interrupt | Fault signals respond from the power monitoring circuit (boost board) |
| Timer interrupt | Software timing and pulse width modulation output |
| Comparator | External or internal power supply determination |
| I2C | Power information reading |
| UART | ARM® board communication |

Based on that, MSP430G2553 microcontroller was selected.

### 3.3.2 DC-DC conversion circuit

The DC-DC conversion circuit is used to convert the unstable input voltage, usually 9-16 V using lead batteries, to a stable 5 V/3.3 V/1.1 V voltage for specific internal power requirements. Considering the large conversion voltage drop, the PTH08080W switching mode power supply module is used since it has a conversion efficiency of 80%-93% in the entire

load range, which is 4.5-18 V for input, and 0.9-5.5 V for adjustable output.

### 3.4 Design of ARM® main control board

The ARM® board integrates the core board of the AM4379 (Texas Instrument, 2013) processor and various communication interfaces. The structure of main control board is shown in Figure 4.

The ARM® board communicates with FPGA using General Purpose Memory Controller (GPMC) interface and interacts

with the power board through the UART serial port. Acquisition data is stored by micro-SD card. Remote monitoring and control are carried out through NB-IoT. In addition, 2.4 GHz and 5 GHz dual-band Wi-Fi module is introduced to realize wireless control and data transmission. For back-up data transmission and debugging interfaces, ethernet and RS232 serial ports are also reserved. In this case, an industrial-grade core board SOM-TL4379 based on TI AM4379 is used.

NB-IoT realizes remote data transmission between PSU and upper computer in low power consumption. The NB-IoT chip

named M5310-A communication module supports LTE Cat-NB1/NB2 and has the characteristics of low power consumption (3 µA @ PSM mode) and ultra-high operating temperature range. Data is transmitted from MCU through UART interface to the NB-IoT module, then uploaded to the OneNET cloud platform via the base station (China mobile, 2018).

Wi-Fi, on the other hand, is introduced for high-speed wireless acquisition data uploading (to upper computer) and other functions. In the proposed instrument, TiWi5 Wi-Fi module supporting 2.4 GHz and 5 GHz dual frequency bands is used

(Laired Connectivity, 2016). Comparing to 2.4 GHz, 5 GHz frequency band relieves network congestion, providing better signal coverage and faster transmission speed.



### 3.5 Design of FPGA board

The FPGA board is used to process a huge number of digital signals and logic control (De La Piedra A. and Braeken A. Touhafi, 2012). The main functions are as follows:

1) Communicate with the acquisition stations based on the improved LVDS interface to achieve the acquisition station control and data uploading from acquisition stations to the PSU.

2) Receive control commands from the ARM® board and transmit the acquired data to the ARM® board using the GPMC interface.

3) Control the clock board, and complete GPS signal analysis as well as Oven Controlled Crystal Oscillator (OCXO) 150 calibration.

LVDS communication is well supported by FPGA 5CEBA2F7 with integrated LVDS transceiver on-chip. OCT (On-Chip Termination) can be used by LVDS receiver to achieve impedance matching, without external termination resistors.

The transmission signal on the line is a high-voltage signal based on Power on Ethernet (PoE) technology realizing power supply and data transmission simultaneously. And this gives the explanation of the ARM® + FPGA dual core design. LVDS 155 data in Manchester encoding from cable are decoded by FPGA and transmitted to ARM® board then for further data uploading, processing, or internal data storage.

### 3.5 Design of clock board

To improve the measurement accuracy of ACs and achieve synchronization between PSUs, an accurate clock source is 160 required (Cao et al., 2010). The high-precision clock system design scheme is shown in Figure 5. FPGA uses 10 accumulative Pulse Per Second (PPS) signal generated by the GPS receiver as the time gate to measure the frequency of the OCXO. Frequency deviation of the OCXO is then calculated and the corresponding calibration value is obtained. According to the calibration value, the digital voltage control value is generated and sent to the Digital to Analog Converter (DAC). After conversion, the analog voltage is output to the voltage control terminal of the OCXO to realize the calibration.

LEA-6T GPS module, whose timing accuracy reaches 15 ns, perform timing when only one satellite is visible is utilized in this case (Ublox, 2017), to unsure the precision of PPS source.

The OCXO on the other hand reduces the output frequency change of oscillator caused by temperature variation by stabilizing the internal temperature, but there is still cumulative error. As the working time increases, the clock error will gradually increase. Take the OCXO in this case as an example, whose output frequency is 12.288 MHz with frequency 170 stability is $\pm 5 \times 10^{-8}$. If no calibration is implemented, the accumulated error after one day could be:


$$\Delta t = (1 - \frac{f_s}{f_s \pm f_s \cdot k}) \times 3600s \times 24$$

$$= k \cdot \frac{1}{1 \pm k} \cdot 86400$$

$$= 5 \times 10^{-8} \times \frac{1}{1 \pm 5 \times 10^{-8}} \times 86400s \approx 4.32 \times 10^{-3} s = 4.32ms$$

(1)

Here, $f_s$ is the standard frequency of OCXO and $k$ represents the frequency stability.

Assuming that the sampling interval is set to 1ms, there will be an error of 4 sampling points after a one-day acquisition. Therefore, the OCXO must be calibrated to improve the frequency accuracy.

The selected OCXO is voltage controlled, thus a voltage of 0-5 V is applied to the CONTROL pin to achieve precise adjustment of the output frequency and the pull range is $\pm 5 \times 10^{-7}$ s. The 12bit DAC7512 (Texas Instrument, 2012) is used to produce the control voltage, producing the frequency adjustment of $2.5 \times 10^{-10}$ s per step. Since that, accurate frequency calibration could be achieved.

## 4 Design of software program

The ARM® board applications are developed based on the Linux platform. The main tasks are: 1) communicating with the power board to manage the PSU and the acquisition station arrangement; 2) accessing the micro SD card for file storage; 3) communicating with the FPGA board to control the acquisition stations and to receive acquired data from the acquisition stations; 4) accessing the Wi-Fi module to realize the wireless communication function; 5) communicating with the NB-IoT module to realize remote control and real-time quality monitoring; 6) debugging through serial port and network port.

Each part of the function is implemented by a separated thread to improve the response speed of the applications and ensure that the system would not be blocked by a single task. In addition, there is a main thread module running after system launch, which is mainly responsible for device initialization and starting other threads.

### 4.1 Power board programming

The power board mainly realizes the functions of power monitoring, fault response, boost board control, interface board 190 control, etc. The program flow chart is shown in Figure 6.

After the MCU on the power board is powered on, modules including GPIO, timer, serial port, comparator, I2C interface, etc. will be initialized firstly. Then, the global interrupt is turned on, while the MCU enters the low-power-consumption mode and will not be awakened until interrupt is generated.

The timer interrupt interval is set to examine the power source status. When timer interrupt request is generated, the MCU 195 will read the data in the register of LTC2945. If the threshold value is not exceeded, the data will be uploaded directly to the ARM® board in specific format, and the MCU sleeps again to enter the low power consumption mode. If there is a certain item of data that exceeds the threshold, further judgment will be executed to analyze the type of data that exceeds the


threshold. When the battery voltage drops to a certain segment point, the status of the battery indicator will be switched. If the output voltage, current, or system power exceeds the normal operating range, an early warning signal will be issued to

prompt the user to troubleshoot.

There are three interrupt sources for I/O interrupts, connecting to the "Alert" signals of three pieces of LTC2945 respectively. If the fault comes from the boost board, the MCU will shut down the corresponding boost module firstly to ensure the safety of the acquisition line, then send a fault signal through the indicator light to remind the user and upload the fault data to the ARM® main control board. If the fault comes from the battery that input power, since the power switch

cannot be turned off directly by software, the MCU can only issue a fault indication and upload fault data.

The serial port of the MCU is connected to the ARM® board, and the serial port interrupt is used to respond to the control commands of the ARM® board. After receiving the serial port data, the MCU will parse out the command and determine whether it conflicts with the current state. If no conflict, the corresponding operation will be executed directly. However, on condition that a conflict exist, meaning that there is a communication problem between the power board and the ARM® main

control board, the conflict will be reported to the ARM® board.

The power selection indication signal output by LTC4416 is connected to the comparator input port of MCU. When the power supply is switched between the external power supply and the back-up battery, the comparator interrupt is triggered. Then the MCU will determine the current power source and report the status to the ARM® board.

### 4.2 ARM® application programming

As the main control unit of the PSU, the ARM® board needs to receive commands from host computer to complete tasks including power supply management, parameter setting as well as storing and uploading data. The amount of data to process is huge and multiple tasks are executed concurrently. Therefore, an operating system must be equipped to manage system resources and schedule multiple applications uniformly, and to provide users with file systems and hardware access interfaces.

The ARM® board applications are developed based on the Linux platform, and details are given in Figure 7. The main tasks are: 1) communicating with the power board to manage the PSU and the acquisition station arrangement; 2) accessing the micro SD card for file access; 3) communicating with the FPGA board to control the acquisition stations and to receive acquired data from the acquisition stations; 4) accessing the Wi-Fi module to realize the wireless communication function; 5) communicating with the NB-IoT module through the serial port to realize remote control and real-time quality monitoring;

6) debugging through serial port and network port.

Each part of the function is implemented by a separated thread to improves the response speed of the application and ensures that the system would not be blocked by a task occupied too much time. In addition, there is a main thread module, running after the system is launched, is mainly responsible for device initialization and starting other threads.



### 4.3 Design of FPGA board program

**4.3.1 Manchester encoding**

The high-speed data transmission between PSU and ACs is based on improved LVDS, and the PoE power supply as well as signal isolation are realized through the isolation transformer. But at the same time, the isolation transformer will also cut off the DC component of the signal. Therefore, only the AC component of the input signal will be observed at the receiver. When continuous binary 0 or 1 appears in the data, the voltage at receiving end will drop, causing signal jitter and bit

errors. Therefore, the Manchester encoding technology is introduced to solve this problem. The positive edge represents data 0, and the negative edge represents data 1. However, since two data bits are used to represent one bit of information, the data transmission efficiency is reduced by half.

### 4.3.2 Design of clock board control program

The frequency measurement calibration program is implemented by FPGA using HDL (Hardware Description Language),

which is mainly composed of delay reset module, clock calibration module and DAC control module. The program workflow is illustrated as Figure 5. The delay reset module releases a reset signal 10 seconds after power-on to wait for the internal oven-controlled chamber of the OCXO reach the working temperature. The clock calibration module then measures the output frequency of the OCXO while calculating the calibration amount. The gate time used for measurement is the time interval of 11 consecutive PPS signals (10 s). Within the gate time, the output of OCXO is counted and compared with the

theoretical value to generate a calibration value, which is output to the DAC control module. The DAC control module then generates the control signal and writes it in a certain timing sequence to DAC7512. In this way, a single calibration is executed. When the difference between the counted value and the theoretical value decreases to a certain range, the calibration is completed.

### 5 Performance

After the design and assembly of the instrument, a series of verification experiments were carried out to evaluate the major performance of the instrument.

### 5.1 Output power experiment

Supplying power to the acquisition line on both sides is the key function of the power station unit, thus the load capability of the PSU is an important technical parameter. Firstly, a simulated load is used instead of ACs to evaluate the maximum power

output ability. The average measured output power of one side is:

$$P = 47.36V \times 1.39A \approx 65.83W$$
(2)

According to the results above, the total load capacity of the power station unit can reach 131.66 W.





We then implemented an output power evaluation experiment to illustrate the output power with various load quantities. Acquisition stations of one seismic and one electrical acquisition channel have been chosen as the load, which has a total

power consumption of 0.72 W, and the interval between acquisition stations is 20 m in the experiment. Output curve with load increasing is illustrated as figure 8. This experiment shows that the proposed PSU could supply power to 120 ACs.

### 5.2 Remote control experiment

To upload quality control information and realize the human-computer interaction, the communication experiment based on NB-IoT is carried out to achieve the data transmission from end to cloud. As is shown in Figure 9, the remote-control

functions are including online device scanning, acquisition status control, battery status, GPS information, network signal strength, etc. In the evaluation experiment, PSU supplied power to 8 seismic-electrical hybrid acquisition stations simultaneously, and the acquisition stations are controlled to enter several working conditions through remote commands. By remote controlling, long-term monitoring during joint prospecting of seismic and electrical methods can be carried out, and it has good application prospects in urban underground space detection and shallow surface-wave exploration. Using the

NB-IoT network can avoid serious interference from the wireless network, providing better signal coverage and signal quality.

### 5.3 Comparison

After above-mentioned evaluation experiments, comparison between proposed PSU and the state-of-art LAUL-428 is listed in table 2. First, the proposed PSU owns an advantage in providing power supply to more acquisition stations (AS)

simultaneously than LAUL-428. The test is carried under the condition of 20m interval between each AS with a 2ms sampling interval to match the test condition of LAUL-428 provided by their official manual (Sercel, 2007). Another technical advantage is that PSU has the ability of data communication which is not supported by LAUL-428.

**Table 2.** Comparison between PSU and LAUL-428

|  | **LAUL-428** | **PSU** |
|---|---|---|
| Data transmission speed | 8Mbps, 16Mbps | 16Mbps (Manchester encoding) |
| Maxim acquisition stations | 81 FDUs™ | **120** AS |
| Function | Power supply: 50V output | Power supply: 48V output |
| Operating power | 10.5-15V DC input **2 battery connectors** | **4.5V-18V** DC input **Internal backup battery** |
| Wireless data upload | —— | NB-IoT & Wi-Fi |



| Memory | 30MB local buffer | 32GB microSD card |
|---|---|---|
| Weight | 2.40kg | **2.27kg** |
| Operating Temperature | -40°C to +70°C | -40°C to +70°C |
| Durability & Resistance | **15m deep in water** | Water repellent and dust resistant |

## 6 Conclusion

In this article, a new power station unit is designed for joint geophysical exploration. Combining with hybrid acquisition stations and upper computer software, a novel distributed wireless seismic and electrical method hybrid acquisition system becomes realistic as well.

Compared with the traditional wired seismograph system, the structure of PSU system is much simpler, and the acquisition data could be uploaded directly from PSU to the central unit through wireless local area network.

The proposal of PSU system has several advantages over nodal instruments as well (Qiao et al., 2019). Firstly, power supply through the proposed PSU could be more stable and sustainable and is more convenient for power management. Secondly, unlike the nodal instrument who uploads data individually, the PSU packs data from 120 acquisition stations and upload them to the central unit through WLAN in real-time, therefore alleviate the channel congestion problem, providing a long-term stable work support and a larger channel quantity. Owing to the NB-IoT technology, who gives the functions of remote control and quality check, networking with other IoT based instruments for multi-parameter geophysical exploration could be expected.

The PSU simplifies system structure of seismic and electrical prospecting and reduces the overall cost, forming a wired and wireless hybrid distributed acquisition system. A multitude of geophysical exploration methods, namely reflection, refraction, surface wave exploration, as well as high-density electrical exploration could be carried out. In conclusion, the proposed PSU is a crucial part of a new distributed hybrid acquisition system architecture of seismic and electrical prospecting.

**Acknowledgment**

This work is supported by the Natural Science Foundation of China (No. 41574131 and No. 41204135), the PetroChina Innovation Foundation (No. 2019D-5007-0302).



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





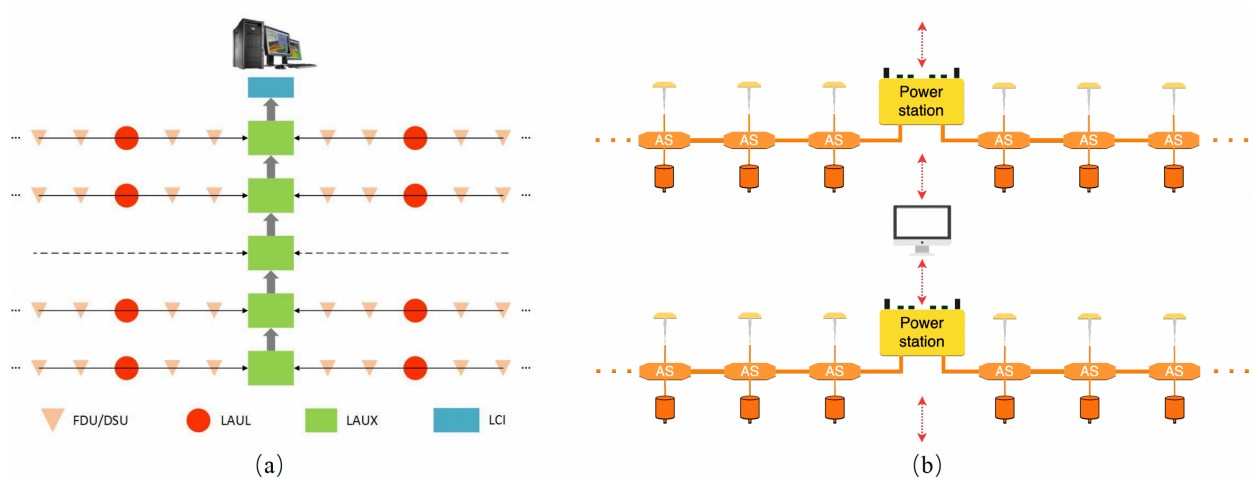

**Figure 1: (a) structure diagram of 428XL system; (b) system structure of PSU (AS: Acquisition Station)**

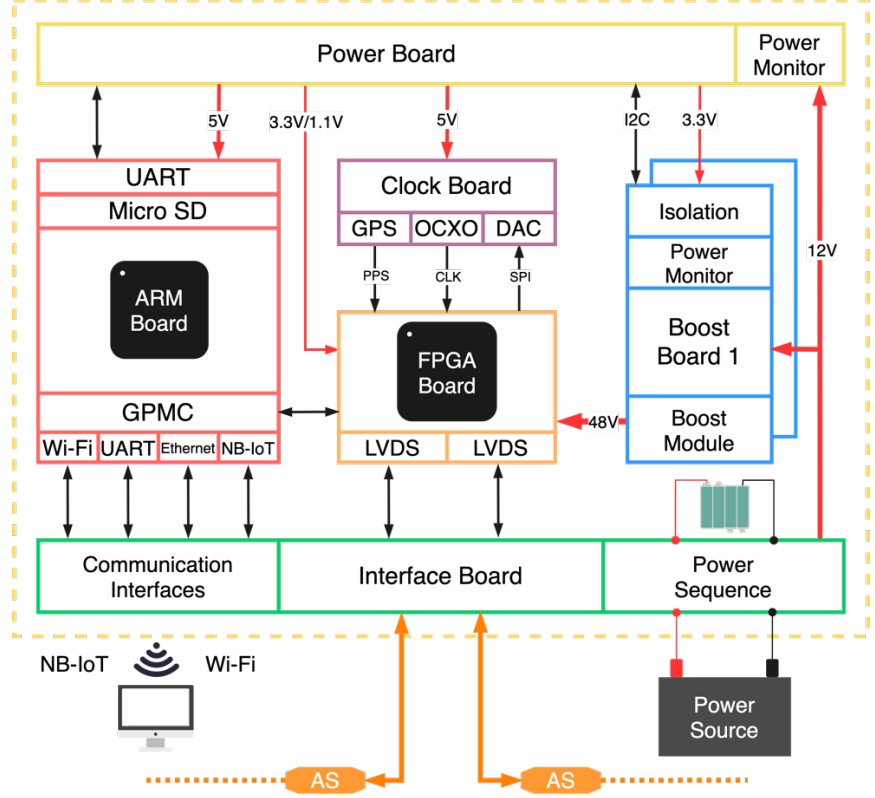

**Figure 2: Hardware structure diagram of PSU.**


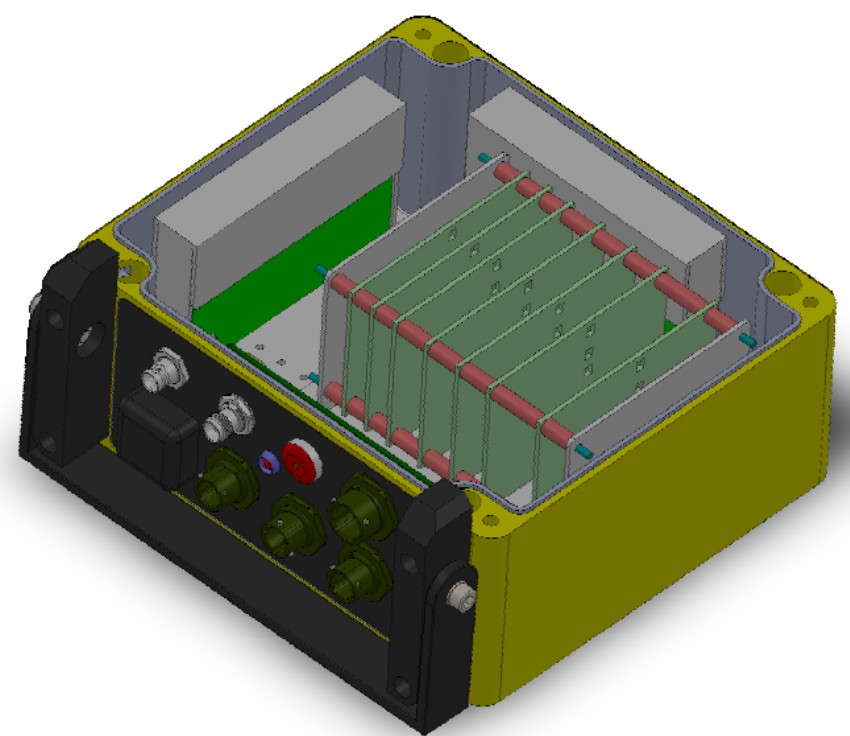

**Figure 3: Three-dimensional structure diagram of PSU's external case.**


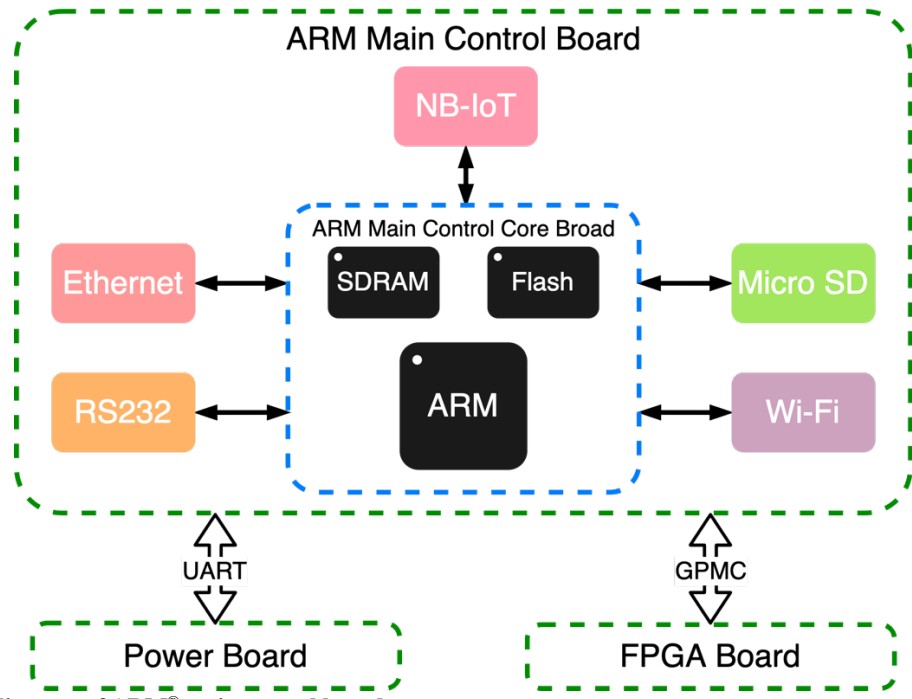

**Figure 4: Structure diagram of ARM® main control board**





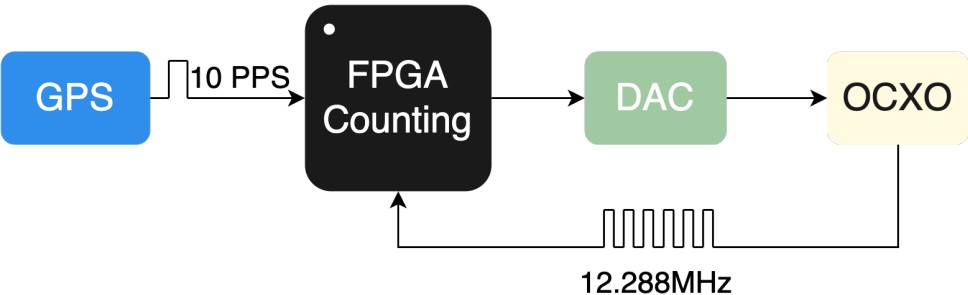

**Figure 5: High-precision clock system design scheme**

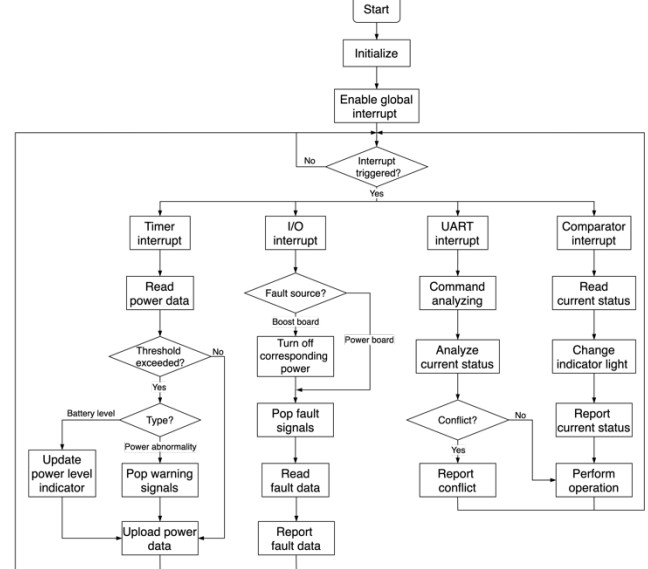


**Figure 6. Power board software program flow chart**

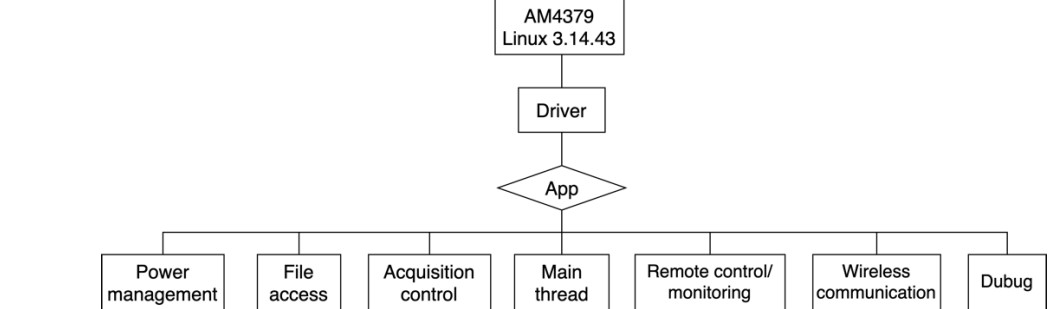

**Figure 7. ARM® main control board application**






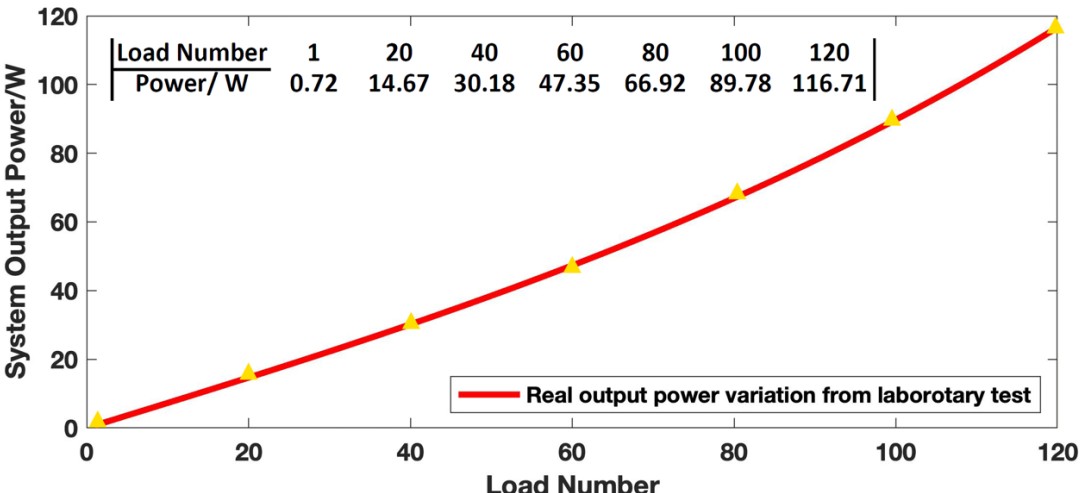

**Figure 8: Output power evaluation test**

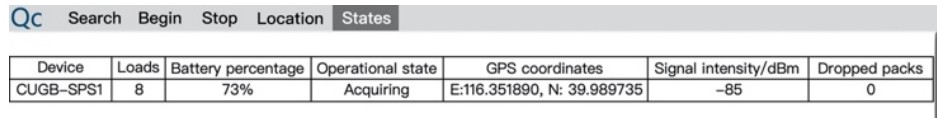

**Figure 9: NB-IoT remote control interface**