# Peer review of "Development of a power station unit in a distributed hybrid acquisition system of seismic and electrical methods based on NB-IoT"

_Geoscientific Instrumentation, Methods and Data Systems, 2022_

## Author Response (AR1)

Editor-in-chief

Geoscientific Instrumentation, Methods and Data Systems

Dear editor and reviewer

Thank you for your warm work and insightful comments and suggestions, we really appreciate your valuable comments which helped us improve the quality of our manuscript (**gi-2022-23,** titled **Development of a power station unit in a distributed hybrid acquisition system of seismic and electrical methods based on NB-IoT**). After careful considerations, we made following revisions according to your comments:

**Anonymous Referee #1**

1) **RC :** Please consider enhancing the abstract part. Instead of the detailed description of the functions achieved, more information such as the reason for this development or the significance of this proposed system should be given.

   **AC :** We simplify the description of functions and enhancing the abstract as suggested. We set our development target from real work with both nodal instrument and conventional instruments, and we found that there's always some problem decreasing the work efficiency when using either system. So we tried to enhance the work efficiency by simplify the system structure while maintaining core features. Therefore, this power station unit is developed. By using the proposed instrument, a new way of networking strategy is given to a bunch of conventional seismic acquisition stations, this also helps to customize a more reasonable acquisition structure.

   **The revised Abstract part is as follows:**

In this paper, we propose a new type of power station unit with wireless data transmission capability. This work breaks the limitation that conventional equipment is unable to upload data directly to central unit. Based on that, a novel distributed geophysical data acquisition architecture is also proposed. enhancing the work efficiency by simplify the system structure while maintaining core features. Designs that realize key functions including isolated high-power output, power management, wireless data transmission as well as high-precision clock synchronization etc. are introduced in this article. The prototype was packaged then, and a series of evaluation experiments were implemented to verify the key parameters of the instrument. Experiment results proved that the overall design of the instrument is feasible, and the key parameters outperforms the industry leading instrument LAUL-428. Due to the wireless networking strategy, the proposed instrument further realizes remote control and real-time data playback through the host computer software, making it suitable for joint geophysical exploration as well as microseismic monitoring. As for system level, it could be customized by connecting different kinds of conventional acquisition stations for many kinds of prospecting targets.

2) **RC:** Please check the grammar of 'weights' on P2 column 47.

   **AC:** Thank you for your comment, this has been corrected.

**The revised part is as follow:**

In fact, as analyzed by Dean et al.(2013), for dense exploration with receiver interval less than 40m, the cabled system weighs less than nodal system, and specifically, when receiver interval is 10 m, the cabled system weighs only 24% of the nodal system.

3) **RC:** The section **3.1 power board control circuit** is not easy to follow, and a clearer way of description is better. Maybe an architecture description before this part

**AC:** Thank you for your comment. This part is revised as below.

**The revised part is as follow:**

The main tasks of power board control circuit are listed as below, therefore an independent microcontroller is needed to achieve those complicated functions. Here, the MSP430G2553 is selected.

4) **RC:** P.6 column 166, please check the usage of 'unsure', which may lead to misunderstanding.

**AC:** Thank you for your comment. The word 'unsure' is corrected as 'ensure'

**The revised part is as follow:**

LEA-6T GPS module, whose timing accuracy reaches 15 ns, perform timing when only one satellite is visible is utilized in this case (Ublox, 2017), to ensure the precision of PPS source.

5) **RC:** Please make sure spaces between all numerals and units to keep a unified format.

**AC:** Thank you for your comment. The format has been revised as below.

**The revised part is as follow:**

**Table 2.** Comparison between PSU and LAUL-428

|  | **LAUL-428** | **PSU** |
|---|---|---|
| Data transmission speed | 8Mbps, 16Mbps | 16Mbps (Manchester encoding) |
| Maxim acquisition stations | 81 FDUs$^{TM}$ | **120** AS |
| Function | Power supply: 50 V output | Power supply: 48 V output |
| Operating power | 10.5-15 V DC input
**2 battery connectors** | **4.5-18 V** DC input
**Internal backup battery** |
| Wireless data upload | —— | NB-IoT & Wi-Fi (2.23 MB/s @ 1km) |
| Memory | 30 MB local buffer | 32 GB microSD card |
| Weight | 2.40 kg | **2.27 kg** |
| Operating Temperature | -40 ℃ to +70 ℃ | -40 ℃ to +70 ℃ |

| | | |
|---|---|---|
| Durability & Resistance | **15m deep in water** | Water repellent and dust resistant |

6) **RC:** Besides NB-IoT remote control, I suggest adding wireless data transmission performance by Wi-Fi.

   **AC:** Thank you for your comment. The data transmission performance of Wi-Fi is 2.23 MB/s at a distance of 1km. To achieve this communication distance, we used directional antenna on central station to amplify the Wi-Fi signal.

   **The revised part is as follow:**

| | LAUL-428 | PSU |
|---|---|---|
| Data transmission speed | 8Mbps, 16Mbps | 16Mbps (Manchester encoding) |
| Maxim acquisition stations | 81 FDUs™ | **120** AS |
| Function | Power supply: 50 V output | Power supply: 48 V output |
| Operating power | 10.5-15 V DC input
**2 battery connectors** | **4.5-18 V** DC input
**Internal backup battery** |
| Wireless data upload | —— | NB-IoT & Wi-Fi (2.23 MB/s @ 1km) |
| Memory | 30 MB local buffer | 32 GB microSD card |
| Weight | 2.40 kg | **2.27 kg** |
| Operating Temperature | -40 ℃ to +70 ℃ | -40 ℃ to +70 ℃ |
| Durability & Resistance | **15m deep in water** | Water repellent and dust resistant |

**Anonymous Referee #2**

1) **RC:** Please make sure spaces between all numerals and units to keep a unified format.

   **AC:** Thank you for your comment. The format has been revised as below.

   **The revised part is as follow:**

Table 2. Comparison between PSU and LAUL-428

| | LAUL-428 | PSU |
|---|---|---|
| Data transmission speed | 8Mbps, 16Mbps | 16Mbps (Manchester encoding) |
| Maxim acquisition stations | 81 FDUs™ | **120** AS |
| Function | Power supply: 50 V output | Power supply: 48 V output |

| Operating power | 10.5-15 V DC input
**2 battery connectors** | **4.5-18 V** DC input
**Internal backup battery** |
|---|---|---|
| Wireless data upload | —— | NB-IoT & Wi-Fi (2.23 MB/s @ 1km) |
| Memory | 30 MB local buffer | 32 GB microSD card |
| Weight | 2.40 kg | **2.27 kg** |
| Operating Temperature | -40 ℃ to +70 ℃ | -40 ℃ to +70 ℃ |
| Durability & Resistance | **15m deep in water** | Water repellent and dust resistant |

2) **RC:** References of Manchester encoding should be given.

**AC:** Thank you for your comment. The references are added as below

**The revised part is as follow:**

**4.3.1 Manchester encoding**

The high-speed data transmission between PSU and ACs is based on improved LVDS, and the PoE power supply as well as signal isolation are realized through the isolation transformer. But at the same time, the isolation transformer will also cut off the DC component of the signal. Therefore, only the AC component of the input signal will be observed at the receiver. When continuous binary 0 or 1 appears in the data, the voltage at receiving end will drop, causing signal jitter and bit errors. Therefore, the Manchester encoding technology is introduced to solve this problem. The positive edge represents data 0, and the negative edge represents data 1 (Lalitha V. and Kathiravan S, 2014; Suchitra S., 2013). However, since two data bits are used to represent one bit of information, the data transmission efficiency is reduced by half.

Lalitha, V., Kathiravan, S., A review of manchester, miller, and fm0 encoding techniques, SmartCR, *4*(6), 481-490, 2014.
Suchitra S., Vhdl Implementation of Manchester Encoder and Decoder, International Journal of Electrical, Electronics and Data Communication, 1 (2) , 2320-2084, 2013.

3) **RC:** The section 3.3.1 is not expressed in a very clear way, and could be improved to make it easy to read.

**AC:** Thank you for your comment. This part is revised as below.

**The revised part is as follow:**

The main tasks of power board control circuit are listed as below, therefore an independent microcontroller is needed to achieve those complicated functions. Here, the MSP430G2553 is selected.

**Anonymous Referee #3**

1) **RC :** Which kind of electrical method is used in the acquisition stations? What are the differences between the seismic AS and electrical AS when interfacing the two links to PSU?

**AC :** Thank you for this comment. Before this PSU has developed, our research team had some development experiences on a cabled acquisition station with 1 seismic channel and 1 electrical channel. In this case, high-density electrical method and seismic joint exploration could be carried out with convenience. Therefore, the connection way is the same.

**The revised part is as follows:**

Combining the seismic and electrical acquisition stations developed by our team, a new type of power station unit (PSU) is proposed, thus a geophysical instrument networking method is also proposed in this article. The point is to retain the original functions of the LAUL and integrate the upward communication and human-computer interaction capabilities of the LCI by wireless communication. Therefore a 120-channel acquisition arrangement is developed based on a single PSU. This arrangement can be further expanded through wireless networking, which could be very convenient for joint geophysical exploration including seismic exploration and high-density electrical method exploration on complex terrain (R. G. Heath, 2008). Compared with traditional wired systems, the proposed structure can reduce a considerable number of equipments to form an efficient work and data transmission flow.

2) **RC:** How is the networking method is designed using NB-Iot in terms of network topology and routing algorithm? What's the data throughput and package latency? Is there any test launched for evaluation of these network performance?

**AC:** Thank you for your insight comment. The topology of this NB-IoT network is shown in the figure. The PSU communicate with NB-IoT base station through internal NB-IoT module, and then communicate with the OneNET cloud platform. Client access data from the OneNET platform. The communication protocol was set to LWM2M.

[Figure]

**Figure 5: NB-IoT based networking topology**

We implemented indoor experiment to verify the network performance of NB-IoT module. The result shows that the downstream data throughput is 8.86 kbps averaged and the package latency is 1183 ms.

**The revised part is as follow:**

**3.4 Design of ARM® main control board**

The ARM® board integrates the core board of the AM4379 (Texas Instrument, 2013) processor and various communication interfaces. The structure of main control board is shown in Figure 4.

The ARM® board communicates with FPGA using General Purpose Memory Controller (GPMC) interface and interacts with the power board through the UART serial port. Acquisition data is stored by micro-SD card. Remote monitoring and control are carried out through NB-IoT. In addition, 2.4 GHz and 5 GHz dual-band Wi-Fi module is introduced to realize wireless control and data transmission. For back-up data transmission and debugging interfaces, ethernet and RS232 serial ports are also reserved. In this case, an industrial-grade core board SOM-TL4379 based on TI AM4379 is used.

NB-IoT realizes remote data transmission between PSU and upper computer in low power consumption. The NB-IoT chip named M5310-A communication module supports LTE Cat-NB1/NB2 and has the characteristics of low power consumption (3 $\mu$A @ PSM mode) and ultra-high operating temperature range. The network topology is shown in Figure 5. Data is transmitted from MCU through UART interface to the NB-IoT module, then uploaded to the OneNET cloud platform via the NB-IoT base station (China mobile, 2018). Client then accesses the uploaded data from the OneNET cloud platform.

**5.2 Remote control experiment**

To upload quality control information and realize the human-computer interaction, the communication experiment based on NB-IoT is carried out to achieve the data transmission from end to cloud. As is shown in Figure 10, the remote-control functions are including online device scanning, acquisition status control, battery status, GPS information, network signal strength, etc. In the evaluation experiment, PSU supplied power to 8 seismic-electrical hybrid acquisition stations simultaneously, and the acquisition stations are controlled to enter several working conditions through remote commands. Other indoor performance experiments have been implemented as well, and the result shows that the downstream data throughput is 8.86 kbps averaged and the package latency is 1183 ms based on LWM2M protocol. By remote controlling, long-term monitoring during joint prospecting of seismic and electrical methods can be carried out, and it has good application prospects in urban underground space detection and shallow surface-wave exploration. Using the NB-IoT network can avoid serious interference from the wireless network, providing better signal coverage and signal quality.

3) **RC:** As for the power test experiment, is the power consumption of cable taken into account? The electrical current of cable segments at different positions will be different due to the AC power consumption, how is the electrical model of the system composed of electrical cable and acquisition stations is constructed?

**AC:** Thank you for your insight comment. In this experiment, we read the output power data from the LTC2945 chip which is the output power of the whole acquisition line, including the power consumption on the cable. As you mentioned, the electrical model in this case would be

complicated due to many aspects, therefore we measured and recorded the actual output power value instead of simulation or calculation to obtain a precise result. To reduce the power consumption on the cable, the PSU is designed to have 2 output ports, one on each side.

**The revised part is as follow:**
We then implemented an output power evaluation experiment with AS to illustrate the output power variation with different AS quantities. Acquisition stations of one seismic and one electrical acquisition channel have been chosen as the load, and the interval between acquisition stations is 20 m in the experiment. All two power output port are used to decrease the power consumption on the cable. Output curve with AS quantity increasing is illustrated as Figure 9, which was fitted from 7 sets of discrete output power value points listed in the table at the upper left corner. This experiment proved that the proposed PSU could supply power to 120 AS.

4) **RC:** Is any field test is launched to validate the field performance?

   **AC:** Thank you for your comment. Unfortunately, there's no big test launched to validate the performance on field, only several separated performance tests shown in the manuscript were implemented. We are also looking forward to an opportunity to test the whole system once on the actual field.

5) **RC:** In line 58, AC is used to denote acquisition station, while AS is used to denote the same in figure 1(b) and figure2. There are some other spell errors the author should pay attention to, such as "carry" in line 275.

   **AC:** Thank you for your comment for those mistakes. We unified the abbreviation of Acquisition Station by AS. And several spelling errors as well as the format mistakes had also been corrected.

   **The revised part is as follow:**
   The proposed PSU based system is shown in Figure 1(b). A single PSU could connect and supply power to a series of acquisition stations (AS) via the cable by two connectors, one on each side. The PSU serves as the main control unit by sending the control commands to the AS in a relay manner. After signal acquisition, the data are relayed on the line to upload from the AS at the end of the arrangement. The intermediate AS add their own data to the data frame and then continue to upload forward until the PSU. Meanwhile, the PSU also performs as the human-computer interaction interface. Unlike the 428XL system, no cables or additional LCI are required between PSU and the central control unit. Instead, Wi-Fi and Narrow Band Internet of Things (NB-IoT) are implemented as remote data transmission techniques, thus expanding the system scale to a larger extent through wireless networking. This layout is flexible and suitable for complex terrains in the wild or urban underground space detection.

   **5 Performance**
   After the design and assembly of the instrument, a series of verification experiments were implemented to evaluate the major performance of the instrument.

We tried our best to improve the manuscript, and we really appreciate for Editors & Reviewers' warm work earnestly, and hope that the correction will meet with approval.

Once again, thank you very much for your comments and suggestions.

Feng GUO, Qisheng ZHANG, Shenghui LIU
China University of Geosciences Beijing
zqs@cugb.edu.cn